# Genome-Wide Identification and Functional Analysis of the Calcineurin B-like Protein and Calcineurin B-like Protein-Interacting Protein Kinase Gene Families in Chinese Cabbage (*Brassica rapa* ssp. *pekinensis*)

**DOI:** 10.3390/genes13050795

**Published:** 2022-04-28

**Authors:** Qianwen Wang, Kai Zhao, Yuqiang Gong, Yunqiang Yang, Yanling Yue

**Affiliations:** 1College of Landscape and Horticulture, Yunnan Agricultural University, Kunming 650201, China; wqw18337387963@163.com (Q.W.); kailixian1023@aliyun.com (K.Z.); gyq415@126.com (Y.G.); 2The Germplasm Bank of Wild Species, Kunming Institute of Botany, Chinese Academy of Sciences, Kunming 650201, China; yangyunqiang@mail.kib.ac.cn

**Keywords:** abiotic stresses, *BraCBL*–*BraCIPK*, Chinese cabbage, expression profiles, functional differentiation, preferential interactions

## Abstract

In plants, calcineurin B-like proteins (CBL) are a unique set of calcium sensors that decode calcium signals by activating a plant-specific protein kinase family called CBL-interacting protein kinases (CIPKs). The CBL–CIPK family and its interacting complexes regulate plant responses to various environmental stimuli. Chinese cabbage (*Brassica rapa* ssp. *pekinensis*) is an important vegetable crop in Asia; however, there are no reports on the role of the CBLs–CIPKs’ signaling system in response to abiotic stress during cabbage growth. In this study, 18 CBL genes and 47 CIPK genes were identified from the Chinese cabbage genome. Expansion of the gene families was mainly due to tandem repeats and segmental duplication. An analysis of gene expression patterns showed that different duplicate genes exhibited different expression patterns in response to treatment with Mg^2+^, K^+^, and low temperature. In addition, differences in the structural domain sequences of NAF/FISL and interaction profiles in yeast two-hybrid assays suggested a functional divergence of the duplicate genes during the long-term evolution of Chinese cabbage, a result further validated by potassium deficiency treatment using trans-*BraCIPK23.1/23.2/23.3 Arabidopsis thaliana*. Our results provide a basis for studies related to the functional divergence of duplicate genes and in-depth studies of *BraCBL*–*BraCIPK* functions in Chinese cabbage.

## 1. Introduction

During growth and development, plants are exposed to a variety of stresses, such as drought, salinity, cold, K^+^ deficiency, pests, and diseases [1]. In response to numerous environmental stimuli, plants induce and regulate a range of biochemical reactions through a complete set of signaling systems. This complex system can sense, respond to, and transduce stress signals at cellular and molecular levels [2]. Ca^2+^, one of the second messengers, is widely present in cells and regulates a variety of growth and developmental processes in plants, as well as responses to abiotic and biotic stresses [3,4]. In the calcium regulatory network, Ca^2+^ receptors receive calcium ion signals and activate the phosphorylation of downstream proteins or directly interact with downstream proteins, ultimately triggering a series of intracellular biochemical reactions [5].

The calcineurin B-like protein (CBL)-CBL-interacting protein kinase (CIPK) system is involved in the regulation of plant responses to abiotic stress by sensing and decoding Ca^2+^ signals through phosphorylation. CBLs, also known as SOS3−like calcium binding proteins (SCaBPs), are a unique class of Ca^2+^-sensing proteins that contain EF-hand calcium-binding domains in the protein structure [6]. CIPK consists of two structural domains, a conserved N-terminal kinase catalytic domain containing a phosphorylation site activation loop, and a C-terminal regulatory domain with an NAF/FISL motif and a highly distinct protein phosphatase interaction (PPI) motif [6,7]. The NAF/FISL motif of the C-terminal regulatory domain contains a 21-amino acid fragment consisting of the highly conserved N, A, and F (NAF) or F, I, S and L (FISL), which interacts with CBL and suppresses the self-inhibition, allowing the substrate to bind to the structural domain of the kinase [8]. The CBL protein then binds to the NAF/FISL domain at the C-terminus of CIPKs [9], resulting in formation of the CBL–CIPK complex.

CBL and CIPK genes have been identified in many species, such as *A. thaliana* [10], rice [11], sorghum [12], *Chlorella vulgaris* [12], grape [13], pear [13], soybean [14], oilseed rape [15], turnip [16], and maize [17], among others. These studies have extended the analysis of CBL–CIPK interactions to the entire families of CBLs and CIPKs to uncover their functions. Previous studies have reported that the CBL–CIPK complex is involved in mediating Ca^2+^ signaling induced by various stresses, such as salt tolerance regulation (AtCBL4/AtCBL10-AtCIPK24 pathway and TaCBL3-TaCIPK29 pathway) [18,19,20], low potassium regulation (AtCBL1/9-AtCIPK23 pathway, OsCBL1-OsCIPK23 pathway, and VvCBL2-VvCIPK3 pathway) [21,22,23,24], nitrogen regulation (AtCBL1/9-AtCIPK23 pathway) [25], high magnesium regulation (AtCBL2 and AtCBL3 with 4 AtCIPK3/9/23/26) [26,27], low temperature regulation (AtCBL1-AtCIPK7 pathway) [28], drought regulation (AtCBL1/9-AtCIPK23 pathway) [29], pH stress (AtCBL2-AtCIPK11 pathway) [30] and abscisic acid (ABA) regulation (AtCBL9-AtCIPK3 pathway and AtCBL1-AtCIPK1 pathway) (Figure 1) [31,32]. Studies on the CBL–CIPK network have extensively demonstrated the interactions, specificity, and overlap between various members of the CBL and CIPK families. These findings have reflected the functional specificity and redundancy of the CBL and CIPK genes. For example, CBL1 and CBL9 of *A. thaliana* have similar amino acid sequences with overlapping and specific functions. This observation suggested that calcium sensors with high sequence similarity or close phylogenetic relationships might have very different functions and that duplicate genes have often developed novel functions during evolution.

Chinese cabbage (*Brassica rapa* ssp. *pekinensis*) is a vegetable crop of the Brassica family and is one of the most important cash crops in the world, especially in Asia. Chinese cabbage is affected by many unfavorable environmental conditions during growth and development, including salinity stress, waterlogging stress, heavy metal stress, and temperature stress, which can lead to a series of changes in the cabbage [33]. The CBL–CIPK system plays an important role in plant responses to abiotic stresses [34,35,36], and some members of the CBL and CIPK families have been investigated in several species [11,12,13,14,15,16,17,34,36]. To date, no systematic analysis has reported the mechanism through which the CBL–CIPK complex is involved in the growth and development of Chinese cabbage and the response to abiotic stresses. In this study, a genome-wide analysis of cabbage was performed and 18 *BraCBL* genes, as well as 47 *BraCIPK* genes, were identified. The genomic information, phylogenetic relationships, and chromosomal localization were then analyzed. The expression analysis after low temperature, ABA, K^+^, Mg^2+^, PH, and NaCl stress treatments, as well as the interactions between the three duplicate genes of BraCIPK23 and BraCBL, were further investigated. Moreover, the phenotypic changes of trans-*CIPK23.1*, *CIPK23.2*, and *CIPK23.3 A. thaliana*, as well as wild-type *A. thaliana*, during potassium deficiency were investigated. Chlorophyll in the leaves and root length were measured. Finally, functional differences between duplicate genes were determined in two gene and protein families. These findings will provide a foundation for further elucidation of how the CBL–CIPK network in Chinese cabbage integrates plant environmental signals and responds to different adversities.

## 2. Materials and Methods

### 2.1. Identification and Structure Analyses of the CBL and CIPK Gene Family in Chinese Cabbage

The 10 *A. thaliana* CBL and 26 CIPK genes were downloaded from TAIR (http://www.arabidopsis.org) (accessed on 19 February 2021) as queries to search against *B. rapa* genomes (http://brassicadb.org/brad/datasets/pub/BrassicaceaeGenome/Brassica_rapa/V3.0/) (accessed on 19 February 2021). Each protein with its domains and functional sites was examined with SMART (http://smart.embl-heidelberg.de/) (accessed on 19 February 2021). All CBL protein sequences containing the EF-hand calcium-binding domains (PS50222), as well as all CIPK protein sequences with protein kinase domains (PS50011) and the NAF/FISL motif (PS50816), were extracted as candidates.

The GenBank non-redundant protein database was used to search against the candidates. DNAMAN software (LynnonBiosoft, San Ramon, CA, USA) was used for the homology analysis between *B. rapa* and *A. thaliana*. Physicochemical parameters including the MW, theoretical PI, grand average of hydropathicity, and the number of amino acids were calculated using the ProtParam tool of ExPaSy [37] (http://web.expasy.org/protparam/) (accessed on 21 February 2021). Putative EF-hand was predicated using a simple modular architecture research tool (http://smart.embl-heidelberg.de/) (accessed on 19 February 2021). Myristoylaton and palmitoylation motifs were predicted using PlantsP (https://mendel.imp.ac.at/myristate/SUPLpredictor.htm) (accessed on 19 February 2021) and CSS-Palm 4.0 software [38], respectively. The diagram of the intron/exon structures of *BraCBL* and *BraCIPK* was analyzed using the TBtools [39]. Subsequently, the MEME program was used to search for conserved motifs in the Chinese cabbage BraCBL and BraCIPK protein sequences [40].

### 2.2. Phylogenetic Analysis and Chromosomal Location

The BraCBL and BraCIPK protein sequences were aligned using the MAFFT version 7 program, and phylogenetic trees were constructed using the MEGA7.0 software with the neighbor−joining method and the 1000 bootstrap test replicates [41]. To map the locations of *BraCBL* and *BraCIPK* genes in Chinese cabbage, the chromosomal distribution of Chinese cabbage genomic sequences was generated by MapChart2.32 software [42]. The synonymous (Ks) and non-synonymous (Ka) substitution rates were estimated with the CodeML program of the PAML4 package [43]. The divergence time (T) of the *BraCBL* and *BraCIPK* gene pair was calculated as *T* = *Ks*/2*λ* (divergence rate of *A. thaliana λ* = 1.5 × 10^−8^) [44].

### 2.3. Plant Material, Growth Condition, and Stress Treatments

Seeds of Chinese cabbage were grown in soil pots and 1/4 Hoagland’s nutrient solution (pH 5.5) under controlled conditions (28 °C day/25 °C night cycle, relative humidity of 75−80%, 200 mmol photons m^−2^ s^−1^ light intensity). The seeds of Chinese Cabbage were qiangshi-17, which was selected by the laboratory. It is a regular variety. For Mg^2+^, K^+^, ABA, PH, high salinity, and cold stress, the seeding was exposed to Mg^2+^ (10 mM) [45], potassium deficiency, ABA (0.2 μM, 1 μM) [46], NaCl (100 mM) [19], PH (8.0) [30], and a cold temperature (4 °C). Each type of stress was treated with 20 nutrient bowls, and each bowl was seeded with about 10 seedlings (*n* = 200). After 10 days of germination, the seedlings were treated with stress and root samples were taken. The roots of seedlings were harvested after treatment for 0.5 h and 1 h, with 0 h as the control. At least three biological and technical repeats were performed for each treatment and taking samples. All samples were immediately frozen in liquid nitrogen and then stored at −80 °C for RNA extraction.

### 2.4. RNA Isolation and Quantitative Real-Time PCR (qRT-PCR) Analysis

Total RNA samples were isolated using the Eastep^®^ Super Total RNA Extraction Kit (Promega, Madison, WI, USA). RNA was quantified by NanoDrop1000 (NanoDrop Technologies, Inc, Wilmington, DE, USA.) with integrity checked on 0.8% agarose gel. Approximately 5 µg RNA was reverse transcribed using the GoScript Reverse Transcription System (Promega, Madison, WI, USA) to generate cDNA. qRT-PCR was conducted in triplicate with different cDNAs synthesized from three biological replicates of different treatments using FastStart Universal SYBR Green Master (Rox, Roche, Indianapolis, IN, USA) and a 7500 Sequence Detection System (Applied Biosystems, Waltham, MA, USA). The reaction parameters for thermal cycling were as follows: 95 °C for 10 min, followed by 40 cycles of 94 °C for 5 s, and 60 °C for 15 s. *B. rapa* tubulin β-2 chain-like (LOC103873913) was amplified as an internal control. The primers used for qRT-PCR are listed in Appendix A. The relative gene expression levels were obtained by dividing the extrapolated transcript levels of the target genes by the levels of the internal control from the same sample. The results were obtained from a comparison of the treatment with the control using independent-samples *t*-test. Statistical analysis was performed using the software IBM SPSS Statistics 25.0 [47].

### 2.5. Yeast Two-Hybrid Assay

Yeast two-hybrid assays were performed using the MatchMaker Y2H system (http://www.clontech.com/) (accessed on 13 April 2021). The coding DNA sequences (CDSs) of *BraCBL* and *BraCIPK* genes were first cloned into the pGADT7 and pGBKT7 vectors, respectively. The *BraCBL* and *BraCIPK* plasmids were then transformed into the yeast strain AH109 according to the method described in the Yeast Protocol Handbook (Clontech; lithium acetate transformation). Transformed yeast cells were grown on the following media: (1) SD-Trp-Leu-dropout medium with deficiency in leucine and tryptophan was used as a positive control for the transformation; (2) SD-Trp-Leu-His-dropout medium with deficiency in leucine, tryptophan, and histidine was used to detect protein interactions under stringent conditions; (3) SD-Trp-Leu-His-Ade-dropout medium with deficiency in leucine, tryptophan, histidine, and adenine was used to detect interactions under stringent conditions. Cell growth was recorded at 48 h intervals over 6 days.

### 2.6. Arabidopsis Treatments and Estimation of the Chlorophyll Content

The assayed plants (overexpression, cipk23.1, cipk23.2, cipk23.3 and WT) were grown on 1/2 solid Murashige and Skoog (MS) for stress analysis. After 4 days, the seedlings were transferred to potassium deficient medium for 10 days [21], the seedling root lengths were measured. The chlorophyll content in the leaf discs floated on potassium deficient was estimated according to the procedure of Arnon (1949) [48]. The leaf discs were homogenized in 1 mL of 80% acetone and the homogenate was centrifuged at 3500× *g*/*n* for 5 min. The supernatant was retained and the absorbance was recorded at 663 and 645 nm. The chlorophyll content was expressed in lg g-1 FW.

## 3. Results

### 3.1. Identification of BraCBL and BraCIPK Genes

In this study, genome-wide identification of *CBL* and *CIPK* in Chinese cabbage was performed (Table 1 and Table 2). Ten CBLs and 26 CIPKs protein sequences from *A. thaliana* were used as queries [45] to search the published genome of Chinese cabbage [49]. Ultimately, 18 CBLs and 47 CIPKs were identified. Of note, *CBL7* was present in *A. thaliana* but not in Chinese cabbage. The CBLs and CIPKs of Chinese cabbage were then named according to their similarity with CBLs and CIPKs of *A. thaliana*, and the sequence similarities to *A. thaliana* AtCBL and AtCIPK were obtained, respectively (Table 1 and Table 2). The results showed that the molecular weights of the CBL protein family members of Chinese cabbage did not differ much from each other, ranging from 194 (*BraCBL5*) to 260 (*BraCBL2.2*) amino acids (except for *BraCBL3.**2*, which had 475 amino acids). All of the proteins contained the EF-hand structures. Except for BraCBL2.2 and BraCBL3.3, which contained two EF-hands, all others contained three EF hands, which provided the structural basis for calcium binding. In addition, all proteins were acidic proteins. BraCBL1.1, BraCBL1.2, BraCBL4.1, BraCBL4.2, BraCBL4.3, BraCBL5, BraCBL9.1, and BraCBL9.2 proteins all contained a conserved N-myristoylation and palmitoylation motif (MGCXXS/T), which could be responsible for membrane localization [50]. The isoelectric point of the identified *BraCIPK* proteins ranged from 5.36 (*BraCIPK17.1*) to 9.24 (*BraCIPK17.3*), and the proteins encoded 274 to 620 amino acids. These variations indicated the diversity of the biochemical features of *BraCIPKs*.

### 3.2. Phylogenetic Relationships and Gene Structure Analysis of BraCBLs and BraCIPKs

To gain insight into the phylogenetic relationships of the *CBL* and *CIPK* families, the complete amino acid sequences of proteins in the CBL and CIPK family were used to construct the respective phylogenetic trees. BraCBL proteins were divided into three subfamilies, namely Group I, Group II, and Group III (Figure 2). To analyze the structural features of *BraCBL*, the gene structures, including exons and introns, were localized based on the genome sequence of Chinese cabbage. With the exception of *BraCBL3.**2*, which had 16 introns, all *BraCBLs* were intron-rich, with family members containing similar intron structures (6–8 introns), suggesting that they originated from the same ancestral gene. The *CIPK* gene family of Chinese cabbage could be divided into two subfamilies based on the presence of introns, namely Group A (intron-rich, 17 *BraCIPKs*) and Group B (no introns, 30 *BraCIPKs*) (Figure 3), which was similar to that reported for other species [10]. Structural differences in *BraCIPK* might allow the *BraCIPK* genes to have different functions, since the structural domains of the functional genes determine the gene function [51].

To analyze the diversity of sequence motifs, the MEME suite was used to further search for conserved motifs in BraCBL and BraCIPK proteins. Fifteen conserved motifs (motif 1–15) in BraCBL and BraCIPK proteins were identified (Appendix A). These motifs might help to predict the gene functions. Further analysis showed that motif 10 of 12 BraCBL proteins contained N-myristoylation sites (Appendix A). For BraCIPK, motifs 1 and 2 were found in the structural domain of protein kinase C (PKC). All BraCIPKs, except BraCIPK4.1, had a protein–protein interaction (PPI) domain (motif 8) (Appendix A) and an NAF/FISL domain (motif 7 or 11). The amino acid residues at sites 2, 3, 4, 7, and 10 of the NAF/FISL motif were quite conserved (Figure 4). Further analysis of the amino acid sequences of the NAF/FISL domains of BraCIPKs (Figure 4) showed that the amino acid sequences of most homologues were consistent, such as in BraCIPK1.1/BraCIPK1.2, BraCIPK6.1/BraCIPK6.2/BraCIPK6.3, BraCIPK7.1/BraCIPK7.2, and BraCIPK13.1/BraCIPK13.2/BraCIPK13.3, among others; however, there were also differences between homologous proteins, such as in BraCIPK4.1/BraCIPK4.2, BraCIPK9.1/BraCIPK9.2, and BraCIPK23.1/BraCIPK23.2/BraCIPK23.3, among others.

### 3.3. Chromosomal Localization and Phylogenetic Analysis of the BraCBL and BraCIPK Gene Families in Chinese Cabbage

To determine the relationship between gene repeats and genetic divergence in the *CBL* or *CIPK* gene families of Chinese cabbage, the chromosomal localization of *BraCBL* and *BraCIPK* was determined (Figure 5). Among these 10 chromosomes, chromosome 1 contained the most of *BraCBL* genes (five genes). There was no gene distribution on chromosomes 4, 5, and 7. Chromosomes 6, 8, and 10 each contained one *BraCBL* gene, whereas the other chromosomes contained 2–4 *BraCBL* genes (Figure 5b). Phylogenetic tree analysis and chromosomal localization analysis revealed six pairs of *BraCBL* as possible duplicate genes, namely, *BraCBL1.1/1.2*, *BraCBL2.1/2.2*, *BraCBL3.1/3.**3*, *BraCBL4.1/4.3*, *BraCBL9.1/9.2*, and *BraCBL10.2/10.3*. In addition, some homologous genes were found to be located on different chromosomes or at different positions on the same chromosome (Figure 5b). Two pairs of homologous genes, *BraCBL3.1/3.**3*, were located at different positions on the same chromosome, whereas other homologous genes were located on different chromosomes. In addition, the distribution of the *BraCIPK* gene was variable on the chromosomes of Chinese cabbage (Figure 5a). Chromosomes 5 and 10 contained the most *BraCIPK* genes and Chromosome 7 contained the least *BraCIPK* gene. Repeats were found in 17 pairs of homologous genes, including 13 gene pairs located on different chromosomes (*BraCIPK1.1/1.2*, *BraCIPK2.2/2.3*, *BraCIPK6.1/6.2*, *BraCIPK7.1/7.2*, *BraCIPK9.1/9.2*, *BraCIPK10.1/10.2*, *BraCIPK12.1/12.2*, *BraCIPK20.1/20.2*, *BraCIPK21.1/21.2*, *BraCIPK22.1/22.2*, *BraCIPK23.1/23.2*, *BraCIPK5/25*, and *BraCIPK26.1/26.2*) and three pairs of duplicate gene pairs located on the same chromosome pairs (*BraCIPK4.1/4.2*, *BraCIPK13.2/13.3*, and *BraCIPK17.1/17.3*) (Figure 5a). The large number of duplications of family members within and between chromosomes could be the main reason for the expansion of the CBL and CIPK gene families in Chinese cabbage.

The divergence times (T) of six pairs of BraCBL proteins were estimated by calculating the Ks and Ka mutation rates and by applying a mutation rate of 1.5 × 10^−8^ per year per synonymous site (Table 3). The results showed that the T of the *CBL* gene in Chinese cabbage ranged from approximately 10.8967 to 16.2833 million years ago (MYA), with a mean divergence time of approximately 13.1467 million years. The Ka/Ks (ω) values for each pair of paralogous genes of *CBL* were calculated. All *CBL* paralogs had ω values less than 1 with a mean value of 0.1462, suggesting that the six pairs of CBL proteins of Chinese cabbage might be subjected to strong purifying selection pressure. Notably, one pair of *CBL* genes of Chinese cabbage (*BraCBL3.1/BraCBL3.**3*, ω = 0.5729) had a high ω value, which suggested that the corresponding genes might have evolved rapidly from a common ancestor. In addition, the divergence time of 17 pairs of *CIPK* paralogous genes of Chinese cabbage was estimated, and it was found that the earliest divergence of the *CIPK* genes in Chinese cabbage was 30 MYA, with a divergence time from 3.4700 to 30.6633 million years and an average divergence time of approximately 15.4755 million years. Interestingly, based on the time of divergence, it was estimated that two pairs of *CIPK* paralogs (*BraCIPK17.1/17.2*, *BraCIPK26.1/26.2*) had recently diverged. Some researchers have proposed that *B. rapa* and *A. thaliana* diverged from the same ancestor approximately 14.5–20.4 MYA [52]. In this study, the divergence times of five pairs of *BraCIPK* paralogs (*BraCIPK10.1/10.2*, *BraCIPK11.1/11.2*, *BraCIPK13.2/13.3*, *BraCIPK22.1/22.2*, and *BraCIPK5/25*; 21.6400–30.6633 million years) was found to have occurred before the origin of Chinese cabbage. The mean of ω for all *BraCIPK* paralogs was 0.1334, which is less than 1. It was speculated that these 17 pairs of *BraCIPK* genes were subject to strong purification selection pressure. In contrast, one pair of *BraCIPK* genes, *BraCIPK26.1/26.2* (ω = 0.5388), had a relatively large ω value, suggesting that they might have evolved rapidly from the previous common ancestor species.

### 3.4. Expression Profiles of BraCBL and BraCIPK Genes after Stress Treatment

To determine the functions of *BraCBL* and *BraCIPK* genes, their expression profiles under different stress conditions were investigated using qRT-PCR (Figure 5), and the possible functional differences in the paralogous genes were also analyzed. The results showed that the expression levels of *BraCBL3.2*, *BraCBL5*, and *BraCBL4.3* were significantly downregulated after the six stress treatments. The transcriptional levels of *BraCBL2.2*, *BraCBL-3.1*, *BraCBL-4.3*, *BraCBL-6*, and *BraCBL10.3* significantly increased with cold treatment compared to those under control conditions. The expression of *BraCBL2.1*, *BraCBL4.1*, and *BraCBL4.2* were significantly induced 1 h after ABA treatment. In addition, potassium treatment significantly induced the expression levels of *BraCBL8* and *BraCBL10.2*. The expression levels of *BraCBL1.1* and *BraCBL1.2* were significantly upregulated after Mg^2+^ treatment, and significant differences were observed in the expression of *BraCBL3.1*, *BraCBL4.1*, and *BraCBL2.1* between the half-hour and one-hour treatment. The expression levels of *BraCBL3.2*, *-5,* and *-3.**3* were downregulated after pH 8.0 treatment. With NaCl treatment, the expression of *BraCBL9.1*, *BraCBL9.2*, *BraCBL1.2*, *BraCBL3.**3*, and *BraCBL8* was upregulated 0.5 h after the treatment, then decreased after 1 h. In addition, the expression profiles of paralogous gene pairs of *BraCBL* were investigated, and different expression patterns were found after different stress treatments. After treatment at 4 °C, *BraCBL3.1* expression was upregulated, whereas *BraCBL3.2* expression was downregulated. After Mg^2+^ treatment, *BraCBL1.1* expression was upregulated, whereas no significant changes were observed for *BraCBL1.2*.

The expression of 47 *BraCIPK* genes was examined after six different stress treatments (Figure 6b), and the results showed that *BraCIPK13.3*, *BraCIPK26.2*, *BraCIPK26.1*, *BraCIPK3*, and *BraCIPK23.1* expression was significantly upregulated after cold treatment, and significant differences were observed in the expression levels of *BraCIPK10.2*, *BraCIPK22.2*, *BraCIPK16*, and *BraCIPK11.2* between the half-hour treatment and 1 h treatment. *BraCIPK11.2*, *BraCIPK13.2*, and *BraCIPK15* expression was upregulated after potassium deficiency treatment. The transcriptional levels of *BraCIPK16*, *BraCIPK5*, and *BraCIPK24* were significantly increased after Mg^2+^ treatment. The transcriptional levels of *BraCIPK14*, *BraCIPK2.3*, and *BrrCIPK13.1* were significantly increased after NaCl treatment; however, the transcriptional levels of *BraCIPK11.1* and *BraCIPK17.3* genes were significantly reduced after the six treatments. Some homologous gene pairs exhibited different expression patterns after various treatments, such as *BraCIPK1.1*/*1.2*, *BraCIPK11.1/11.2*, *BraCIPK13.1/13.2*, and *BraCIPK23.1/23.2/23.3*. The expression of *BraCBL9.1*, *BraCBL9.2,* as well as *BraCIPK14*, *BraCIPK2.3*, and *BraCIPK13.1* was upregulated after salt treatment, whereas the expression of *BraCBL3.2*, *BraCIPK11.1*, and *BraCIPK17.3* was reduced, implying the different functions of different members of *BraCBL* and *BraCIPK* families. The differences in the expression of the homologous gene pairs *BraCBL2.1/2.2*, *BraCIPK1.1/1.2*, *BraCIPK11.1/11.2*, *BraCIPK13.1/13.2*, and *BraCIPK23.1/23.2/23.3* suggested functional divergence of homologous genes.

### 3.5. Analysis of the Interactions between BraCIPK23 Duplicate Genes and BraCBLs Proteins

The interaction between CBL and CIPK depends on the NAF/FISL domain of CIPK [53]. An analysis of the amino acid sequence of the NAF/FISL domain of BraCIPKs revealed differences between homologous genes. To determine whether differences in the NAF/FISL domain affect the interaction with BraCBL, three BraCIPK23 repeat genes BraCIPK23.1, BraCIPK23.2, and BraCIPK23.3 encoding differential amino acid sequences in the NAF/FISL domain were selected for the protein interaction study with BraCBLs. Fifteen BraCBLs were cloned and subjected to yeast two-hybrid (Y2H) systems (Figure 7). The results showed interactions between BraCIPK23.1 and BraCBL2.2, BraCBL4.2, BraCBL9.1, BraCBL9.2; BraCIPK23.2 and BraCBL1.1, BraCBL2.2, BraCBL9.1, BraCBL9.2; BraCIPK23.3 and BraCBL9.2. It is worth noting that whereas BraCBL2.1 and BraCBL2.2 are homologous proteins, BraCBL2.1 did not interact with any of the three duplicate proteins; however, BraCBL2.2 interacted with BraCIPK23.1 and BraCIPK23.2. Furthermore, BraCBL9.1 and BraCBL9.2 are homologous proteins. Whereas BraCBL9.1 interacted with all three duplicate genes, BraCBL9.2 did not interact with BraCIPK23.3. BraCIPK23.1 and BraCIPK23.2 interacted with four BraCBLs, whereas BraCIPK23.3 only interacted with BraCBL9.2. These results suggested that duplicate genes might have different functions.

### 3.6. Phenotypic Observation and Physiological Indicator Measurements of Trans-BraCIPK23 Repeat Genes

Previous studies have shown that CIPK23 plays a regulatory role in low potassium stress [21,22,23]. To further elucidate the functional differences in the duplicate genes, BraCIPK23.1, BraCIPK23.2, and BraCIPK23.3 were transformed into wild-type *A. thaliana* and the phenotypes of the transgenic plants were observed after 10 days of potassium deficiency treatment. The results showed that the trans-CIPK23.1, CIPK23.2, and CIPK23.3 *A. thaliana* strains grew basically the same as the wild-type on normal MS medium, with similar root lengths (Figure 6a,d), and they had basically identical chlorophyll a and chlorophyll b contents (Figure 8b,c); however, in the potassium-deficient medium, yellowing was observed on the leaves of wild-type *A. thaliana* (Figure 8d,e), whereas the three transgenic plants grew normally. Measurements of root length revealed that the root length of trans-CIPK23.1, CIPK23.2, and CIPK23.3 *A. thaliana* strains was increased by 22.02 mm, 28.85 mm, and 13.69 mm, respectively, compared with that of the wild-type. In addition, measurements of chlorophyll content revealed that, compared with those in the wild-type, chlorophyll a content increased by 220.5%, 245.4%, and 118.3%, respectively, and chlorophyll b content increased by 212.1%, 290.4%, and 159.7%, respectively, in the three transgenic plants, which indicated that trans-BraCIPK23.1, BraCIPK23.2, and BraCIPK23.3 plants are more adapted to potassium stress than wild-type *A. thaliana*. Differences in phenotypes and indicators were also observed among the three transgenic plants. Compared with trans-*CIPK23.3 A. thaliana*, trans-*CIPK23.1* and trans-*CIPK23.2* plants showed an 8.33 mm and 15.15 mm increase in roots, 46.8% and 58.2% increase in chlorophyll a content, and a 20.2% and 50.4% increase in chlorophyll b content, respectively. These results indicated that there are indeed differences in the functionality among *BraCIPK23.1*, *BraCIPK23.2*, and *BraCIPK23.3*.

## 4. Discussion

The signaling system consisting of CBLs and CIPKs is one of the key regulatory nodes in various plant signaling pathways during adversity. Comprehensive analyses of CBL and CIPK members were previously performed for many species [12,13,16,54], and studies have shown that CBL and CIPK were formed during early plant evolution; however, there have been no systematic reports on *CBL* and *CIPK* genes in Chinese cabbage. In this study, a genome-wide database search was performed based on the conserved structural domains and the known similarities with *A. thaliana CBL* and *CIPK* sequences to identify 18 *BraCBL* genes and 47 *BraCIPK* genes in the Chinese cabbage genome (Table 1 and Table 2). Both *CBL* and *CIPK* genes were expanded, compared with 10 *CBLs* and 26 *CIPKs* in *A. thaliana*, a phenomenon that also occurred in *Medicago* and cotton [55,56]. Previous studies have shown that *B. rapa* has experienced polyploidization such as γ triploidy (135 MYA) and β (90–100 MYA) and α genome duplications (24–40 MYA) [57]. These three polyploidization events occurred in the evolutionary history of *B. rapa* and have led to chromosome reductions and rearrangements, as well as substantial gene loss, resulting in highly complex gene families that might account for the expansion of the *CBL* and *CIPK* genes of Chinese cabbage. It is noteworthy that *CBL7* is present in *A. thaliana* [10] but not in Chinese cabbage, and the main reason for this was determined as gene loss. These results suggested that some members of the *CBL* family were conserved, whereas others were lost after divergence. There were two or more homologous genes in Chinese cabbage for many of the *CBL* and *CIPK* genes in *A. thaliana*, a result that suggested that expansion of the *CBL* and *CIPK* gene families could have been due to genome duplication [58]. An analysis of the sequences revealed five fragment duplications (27.8%) in 18*BraCBL* genes and 17 fragment duplications (36.2%) in 47 *BraCIPK* genes (Table 3). This finding suggested that fragment duplication primarily promoted expansion of the *BraCBL* and *BraCIPK* gene families.

Expansion of the *CBL* and *CIPK* gene families could lead to functional divergence of homologous genes. In this study, the sequence motifs, as well as the structures of introns and exons of the *CBL* and *CIPK* gene families, were systematically analyzed, and it was found that all *BraCBLs* were intron-rich except *BraCBL3.**2*, which contained 16 introns. The *CIPK* gene family of Chinese cabbage was divided into two subfamilies, namely Group A (intron-rich, including 17 *BraCIPKs*) and Group B (no introns, including 30 *BraCIPKs*, as shown in Figure 3), and a similar pattern of intron-rich/poor CIPK family members was observed in *A. thaliana*, rice, poplar, and soybean [10,59,60,61]. Fifteen conserved sequence motifs (motif 1–15) in BraCBL and BraCIPK proteins were identified using MEME (Figure 4). The analysis showed that motif 10 of 12 of BraCBL proteins contain N-myristoylation sites (Figure 4). All *BraCIPK*s, except *BraCIPK4.1*, were found to contain a PPI domain (motif 8) and an NAF/FISL domain (motif 7 or 11) (Appendix A). Studies have shown that the structural domains of functional genes determine gene functions [52], and it was speculated that structural differences in *BraCIPK* might have allowed *BraCIPK* genes to take on different functions.

The NAF/FISL domain in CIPK is required for the interaction with CBL [52]. The NAF/FISL motif sites are conserved, whereas this study identified differences in NAF/FISL sequences. For example, there were differences in the NAF/FISL motifs between *BraCIPK7.1* and *BraCIPK7.2*, similar to those in *BnaCIPK7* in *rape* [15], and *AtCIPK7* in *A. thaliana* [62]. There were also differences in the amino acid sequences of NAF/FISL between *BraCIPK23.1* and *BraCIPK23.2*, similar to those in *BrrCIPK23.1/23.2* in turnip [16]. Previous studies have found that G15V and T20N point mutations in the OsRacD protein are characterized by binding with different target proteins [63]. Among the interactions, this study found that the duplicate proteins of BraCIPK23.1/BraCIPK23.2 and BraCIPK23.3 interact with four BraCBL proteins and one BraCBL protein, respectively (Figure 6a), and that their interactions with BraCBLs are common yet specific. Previous studies have confirmed that the interaction between *AtCBL1* and *AtCIPK7* might play an important role in the cold stress response [28], whereas the stress response regulated by *AtCBL1* and *AtCIPK1* requires the involvement of ABA [32]. This suggests that these genes might function in different signaling pathways and utilize different interaction partners to deliver signals downstream.

The phenotypes of plants overexpressing *CIPK23.1*, *CIPK23.2*, and *CIPK23.3* were observed in a potassium-deficient environment. Compared with those in the wild-type plants, there was little change in the root length, chlorophyll a content, and chlorophyll b content using the regular MS medium; however, after potassium deficiency treatment, there was a substantial increase in the root length, chlorophyll a content, and chlorophyll b content in these three types of plants. Previous studies confirmed that *AtCBL1/AtCBL9* mediates the localization of *AtCIPK23* to the plasma membrane during low potassium ion stress and that *AtCIPK23* subsequently activates AKT1 through phosphorylation, thus enhancing potassium ion uptake by cells at low potassium ion concentrations [21,64], which was consistent with the results in this study. Moreover, there were differences among the three overexpressing plants, with trans-*CIPK23.1* and trans-*CIPK23.2 A. thaliana* having better root length and chlorophyll content than *CIPK23.3*. The interaction results revealed that BraCIPK23.3 only interacts with BraCBL9.2, which was speculated to be due to the difference in protein interactions. Based on the amino acid sequence differences in the NAF/FISL domains of *BraCIPK23.1/BraCIPK23.2* and *BraCIPK23.3*, it was presumed that locus differences were the main reason for the functional differences among these three genes; therefore, an expansion of the gene families encoding the CBL–CIPK signaling system might have led to novel gene functions through the functional divergence of homologous genes. These findings could provide a reference for other functional studies in this field.

## 5. Conclusions

The CBL–CIPK signaling system is one of the key resistance mechanisms in plant responses to abiotic stress. In this study, 10 CBL and 26 CIPK protein sequences of *A. thaliana* were used to search for CBL and CIPK in Chinese cabbage, and a total of 18 CBL and 47 CIPK were identified. These genes show a high similarity in amino acid sequence, motif compositions, and conservative gene structure. Phylogenetic analysis showed that *BraCBL* and *BraCIPK* were divided into three subfamilies and two subfamilies, respectively. By calculating the time of divergence, these genes may have evolved rapidly from the last common ancestor. In addition, we analyzed the expression profiles of *BraCBL* and *BraCIPK* under different stress treatments and found that different duplicated genes showed different expression trends in Mg^2+^, K^+^, low temperature and other treatments. Especially *BraCBL2.1/2.2*, *BraCIPK1.1/1.2*, and *BraCIPK23.1/23.2/23.3*. Analysis of NAF/FISL domain sequences showed that there were differences between duplicated genes. The yeast two-hybrid test showed that BraCIPK23.1/23.2/23.3 had differentiated functions. In addition, transgenic *A. thaliana* showed that *BraCIPK23.2* performed better under the potassium deficiency treatment. Our results provide reference for further study of functional differentiation of duplicate genes.

## Figures and Tables

**Figure 1 genes-13-00795-f001:**
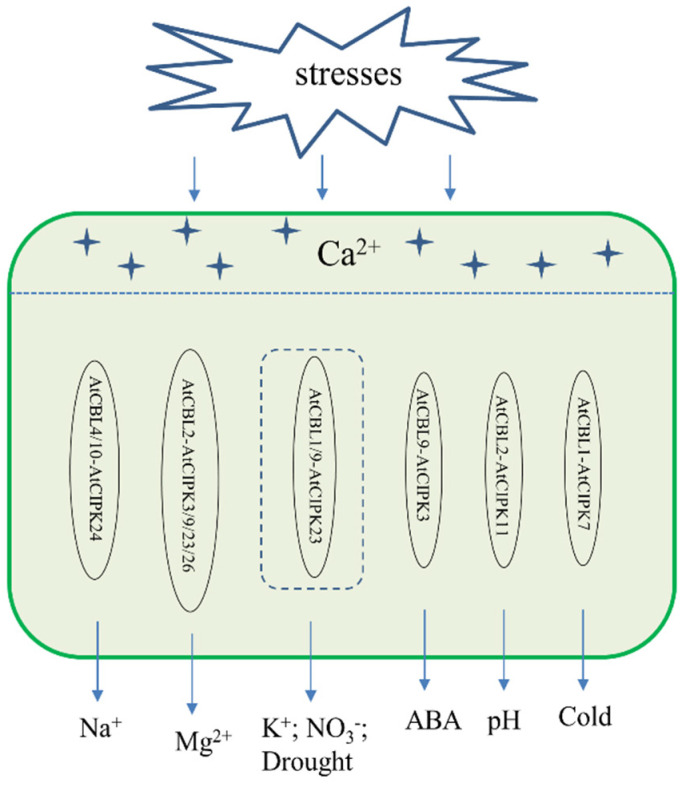
Plant CBL–CIPK signaling system. Plants respond to different stresses through the CBL–CIPK Complex.

**Figure 2 genes-13-00795-f002:**
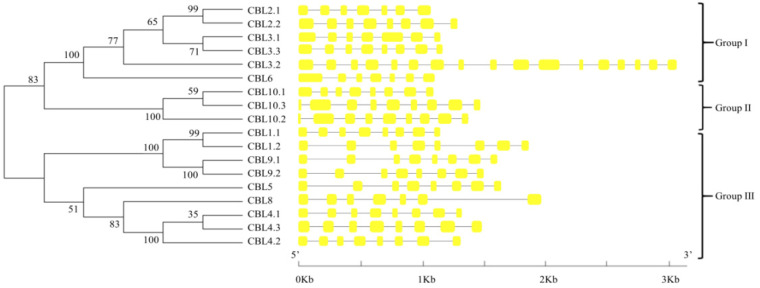
Phylogenetic relationship of Chinese Cabbage CBL proteins. The BraCBL protein sequences were aligned using the MAFFT version 7 program, and phylogenetic trees were constructed using the MEGA 7.0 software with the neighbor-joining method and the 1000 bootstrap test replicates. The tree can be divided into three major clades (Group I–Group III).

**Figure 3 genes-13-00795-f003:**
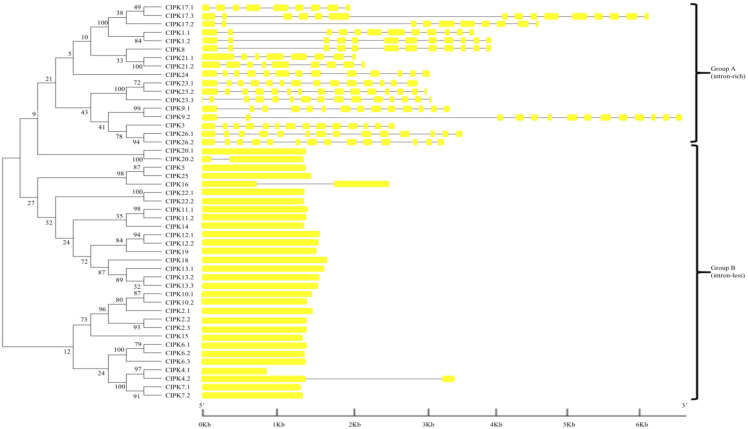
Phylogenetic relationship of Chinese Cabbage CIPK proteins. The BraCIPK protein sequences were aligned using the MAFFT version 7 program, and phylogenetic trees were constructed using the MEGA 7.0 software with the neighbor-joining method and the 1000 bootstrap test replicates. The tree can be divided into three major clades (Group A and Group B).

**Figure 4 genes-13-00795-f004:**
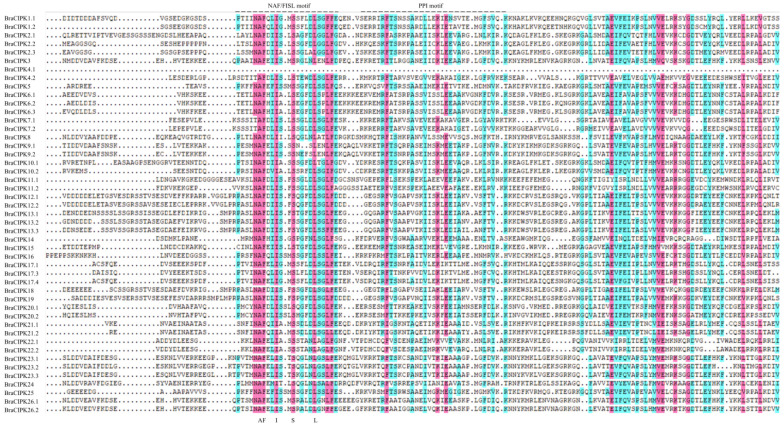
Multiple sequence alignments of 47 *BraCIPKs* amino acids. The conserved NAF/FISL and protein-phosphatase interaction (PPI) motifs of CIPK are marked by dots above the sequence.

**Figure 5 genes-13-00795-f005:**
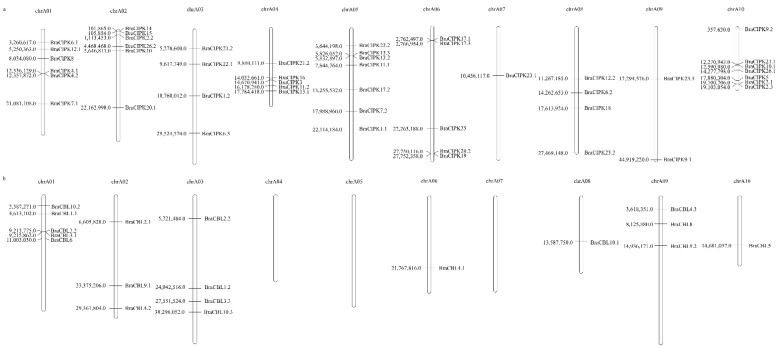
Chinese Cabbage CBLs and CIPKs Chromosome distributions. (**a**) BraCIPKs variably distributed among all on the turnip chromosomes. (**b**) The distribution of BraCBL on 10 chromosomes (Chromosome 4, 5, 7 containing no BrrCBLs).

**Figure 6 genes-13-00795-f006:**
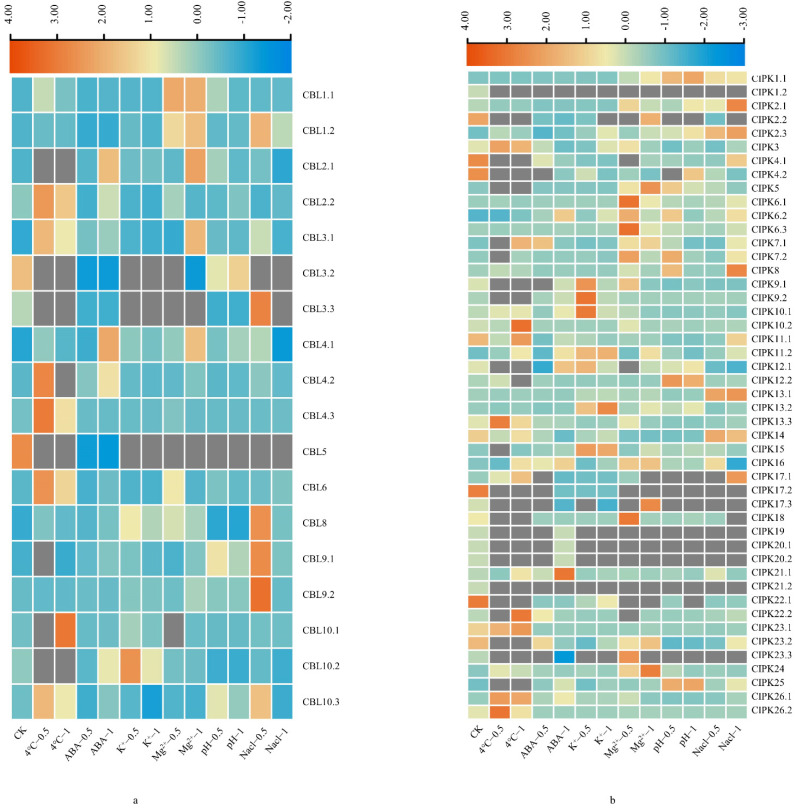
Expression profiles of *BraCBL* and *BraCIPK* genes in different stress treatments. (**a**) Expression analysis of *BraCBLs* in six different treatments at 0.5 h and 1 h. (**b**) Expression analysis of *BraCIPKs* in six different treatments at 0.5 h and 1 h.

**Figure 7 genes-13-00795-f007:**
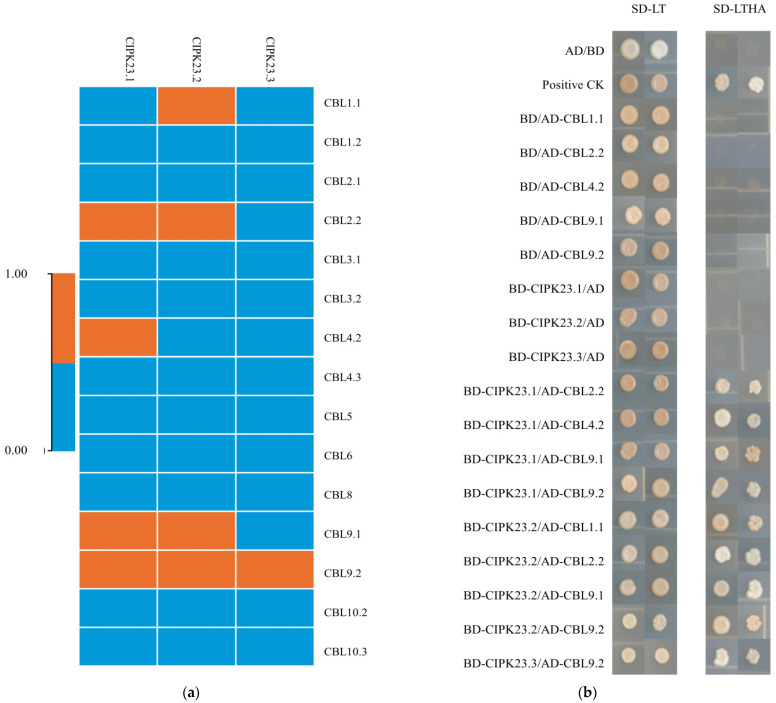
Interaction analysis between the BraCIPK23 repeat gene and BraCBLs protein. (**a**) Heat map summarizing yeast two-hybrid (Y2H) results for all interactions between Chinese Cabbage CBL and CIPK proteins. Orange means interaction, blue means no interaction. (**b**) Yeast two-hybrid experiment showed the interaction between BraCIPK23.1, -23.2, -23.3 and 15 BraCBLs, AD + BD represented negative control, and positive CK represented positive control check.

**Figure 8 genes-13-00795-f008:**
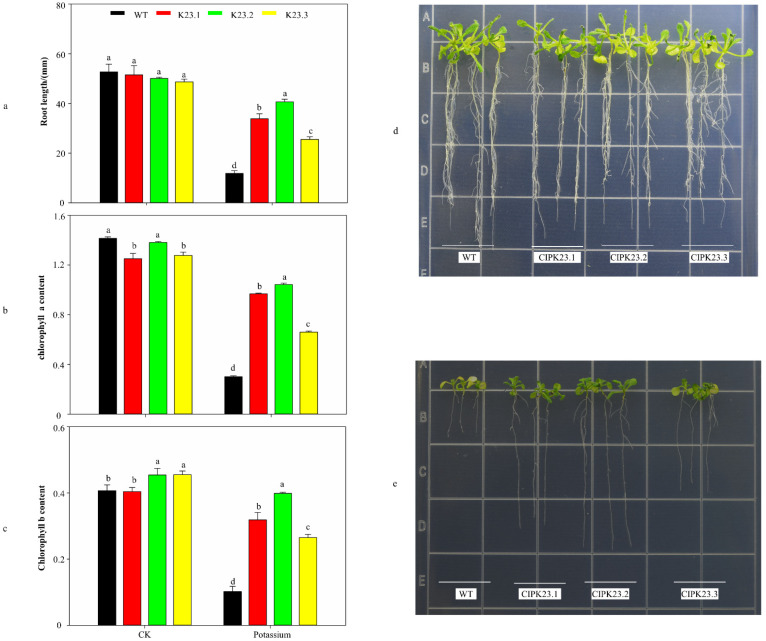
Determination of root length and chlorophyll content and phenotypic observation of trans-*BraCIPK23.1/23.2/23.3 A*. *thaliana* (**a**) Comparison of root lengths of transgenic CIPK23.1, -23.2, -23.3 with wild-type *A. thaliana* on potassium deficient and normal MS mediums. (**b**,**c**) Comparison of chlorophyll a and chlorophyll b content in leaves of transgenic CIPK23.1, -23.2, -23.3, and wild-type *A. thaliana* under potassium deficient and normal MS mediums. (**d**) Comparison of phenotypes of transgenic CIPK23.1, -23.2, -23.3 *A. thaliana* with wild-type *A. thaliana* under normal MS mediums. (**e**) Comparison of phenotypes of transgenic CIPK23.1, -23.2, -23.3 *A. thaliana* with wild-type *A. thaliana* under potassium deficient.

**Table 1 genes-13-00795-t001:** Chinese Cabbage CBL genes identified and their characteristics.

Gene Name	Arabidopsis Ortholog	Identity/%	Gene Locus	MW (Da)	PI	GRAVY	NO. of Amino Acids	No. of EF-Hand	Myristoylaton Motif	Palmitoylation Prediction
*BraCBL1.1*	AtCBL1	92.21	A01:4613102–4615104 (+strand)	24,587.01	4.69	−0.19	213	3	Y	Y
*BraCBL1.2*		91.12	A03:24042516–24044378 (+strand)	24,603.09	4.64	−0.173	213	3	Y	Y
*BraCBL2.1*	AtCBL2	94.57	A02:6605828–6607158 (−strand)	25,865.49	4.89	−0.22	226	3	N	Y
*BraCBL2.2*		80.84	A03:5721464–5722801 (−strand)	29,846.17	5.12	−0.123	260	2	N	Y
*BraCBL3.1*	AtCBL3	89.61	A01:9215862–9217195 (−strand)	25,783.26	4.78	−0.227	226	3	N	Y
*BraCBL3.2*		41.07	A03:27551525–27554719 (−strand)	53,216.02	5.92	−0.249	475	3	N	Y
*BraCBL3.3*		60.78	A01:9213775–9215163 (−strand)	28,005.91	5.27	−0.296	250	2	N	Y
*BraCBL4.1*	AtCBL4	86.85	A06:21767816–21769240 (−strand)	25,583.31	4.97	−0.282	221	3	Y	Y
*BraCBL4.2*		77.28	A02:29361806–29362997 (+strand)	22,451.92	5.04	−0.184	196	3	Y	Y
*BraCBL4.3*		88.19	A09:3618351–3619847 (+strand)	24,986.75	4.95	−0.218	217	3	Y	Y
*BraCBL5*	AtCBL5	80.58	A10:14681036–14683065 (−strand)	22,430.59	5.23	−0.332	194	3	Y	Y
*BraCBL6*	AtCBL6	83.19	A01:11003030–11004304 (+strand)	26,206.29	5.99	−0.21	227	3	N	Y
*BraCBL8*	AtCBL8	89.15	A09:8125880–8127444 (−strand)	24,751.32	5.23	−0.308	214	3	N	Y
*BraCBL9.1*	AtCBL9	91.9	A02:23375206–23376913 (−strand)	24,350.67	4.62	−0.17	213	3	Y	Y
*BraCBL9.2*		90.34	A09:14936171–14937957 (−strand)	24,354.62	4.62	−0.22	213	3	Y	Y
*BraCBL10.1*	AtCBL10	83.33	A08:13587750–13589239 (−strand)	28,462.58	4.82	−0.079	246	3	N	Y
*BraCBL10.2*		72.04	A01:2387271–2388596 (−strand)	24,335.99	4.67	−0.075	211	3	N	Y
*BraCBL10.3*		82.69	A03:30296053–30297485 (+strand)	28,705.94	4.49	−0.001	249	3	N	Y

**Table 2 genes-13-00795-t002:** Chinese Cabbage CIPK genes identified and their characteristics.

Gene Name	Arabidopsis Ortholog	Identity/%	Gene Locus	MW (Da)	PI	GRAVY	NO. of Amino Acids
*BraCIPK1.1*	AtCIPK1	90.83	A05:22114185–22117606 (+strand)	50,008.16	6.08	−0.39	446
*BraCIPK1.2*		86.29	A03:18760011–18763637 (−strand)	47,266.56	8.94	−0.32	420
*BraCIPK2.1*	AtCIPK2	75	A10:19100706–19102103 (+strand)	52,509.11	7.2	−0.425	465
*BraCIPK2.2*		74.4	A02:1113453–1114772 (−strand)	49,660.32	6.78	−0.394	439
*BraCIPK2.3*		73.45	A10:19103054–19104376 (+strand)	49,404.33	9.02	−0.378	440
*BraCIPK3*	AtCIPK3	88.94	A04:14670940–14673364 (−strand)	50,253.49	6.82	−0.534	440
*BraCIPK4.1*	AtCIPK4	50.86	A01:12336129–12336953 (+strand)	30,459.14	9.74	−0.31	274
*BraCIPK4.2*		66.87	A01:12357872–12361057 (+strand)	54,941.5	7.31	−0.171	490
*BraCIPK5*	AtCIPK5	79.29	A10:17880305–17881609 (+strand)	49,526.82	6.14	−0.329	434
*BraCIPK6.1*	AtCIPK6	85.48	A01:3260617–3261942 (−strand)	49,199.57	8.69	−0.289	441
*BraCIPK6.2*		83.79	A08:14262653–14263951 (−strand)	48,461.95	9.04	−0.303	432
*BraCIPK6.3*		80.88	A03:29524568–29525881 (+strand)	49,009.42	8.9	−0.316	437
*BraCIPK7.1*	AtCIPK7	80.47	A01:21081109–21082353 (+strand)	46,300.42	9.21	−0.234	414
*BraCIPK7.2*		79.84	A05:17988961–17990229 (−strand)	47,280.64	8.73	−0.214	422
*BraCIPK8*	AtCIPK8	91.57	A01:8034080–8036979 (+strand)	50,459.22	8.63	−0.215	446
*BraCIPK9.1*	AtCIPK9	90.33	A09:44919220–44922333 (−strand)	49,822.14	8.33	−0.393	445
*BraCIPK9.2*		91.59	A10:357630–363660 (+strand)	50,384.78	8.03	−0.402	447
*BraCIPK10.1*	AtCIPK10	84.36	A10:12590980–12592368 (−strand)	52,751.48	8.02	−0.539	462
*BraCIPK10.2*		75.76	A02:5646811–5648142 (+strand)	50,887.75	8.91	−0.455	443
*BraCIPK11.1*	AtCIPK11	80.27	A05:7844764–7846092 (+strand)	49,700.13	8.29	−0.331	442
*BraCIPK11.2*		79.24	A04:16178280–16179593 (−strand)	49,403.9	8.62	−0.283	437
*BraCIPK12.1*	AtCIPK12	80.98	A01:5250363–5251847 (−strand)	55,311.48	6.75	−0.259	494
*BraCIPK12.2*		83.33	A08:11267185–11268651 (−strand)	55,031.29	6.75	−0.261	488
*BraCIPK13.1*	AtCIPK13	83.72	A04:17784418–17785956 (−strand)	57,880.06	8.17	−0.267	512
*BraCIPK13.2*		80.98	A05:5832897–5834375 (+strand)	55,244.17	8.99	−0.207	492
*BraCIPK13.3*		81.92	A05:5826052–5827512 (+strand)	54,789.57	9.13	−0.221	486
*BraCIPK14*	AtCIPK14	75.05	A02:101865–103160 (+strand)	48,561.02	9.11	−0.241	431
*BraCIPK15*	AtCIPK15	81.41	A02:105854–107125 (−strand)	47,963.47	8.66	−0.37	423
*BraCIPK16*	AtCIPK16	81.99	A04:14032661–14035013 (−strand)	52,367	5.49	−0.399	461
*BraCIPK17.1*	AtCIPK17	70.08	A06:2762497–2764360 (+strand)	39,815.64	5.36	−0.131	357
*BraCIPK17.3*		78.34	A05:13255532–13259898 (+strand)	44,843.86	7.59	−0.211	399
*BraCIPK17.4*		52.18	A06:2766954–2772570 (+strand)	68,620.32	9.24	−0.049	620
*BraCIPK18*	AtCIPK18	83.11	A08:17613924–17615498 (+strand)	58,872	7.54	−0.287	524
*BraCIPK19*	AtCIPK19	81.82	A06:27752359–27753804 (−strand)	54,462.93	9	−0.276	481
*BraCIPK20.1*	AtCIPK20	86.59	A02:22162998–22164305 (−strand)	49,902.91	9.21	−0.45	435
*BraCIPK20.2*		64.27	A06:27750115–27751398 (+strand)	40,057.09	8.87	−0.437	352
*BraCIPK21.1*	AtCIPK21	92.89	A10:12270943–12272878 (−strand)	46,403.52	8.28	−0.166	416
*BraCIPK21.2*		85.13	A03:5278600–5280652 (+strand)	42,800.57	8.59	−0.116	386
*BraCIPK22.1*	AtCIPK22	78.95	A03:9617749–9619041 (−strand)	48,701.31	9.11	−0.362	430
*BraCIPK22.2*		76.68	A05:3644198–3645481 (+strand)	48,342.69	9.15	−0.354	427
*BraCIPK23.1*	AtCIPK23	84.89	A07:10456117–10458829 (−strand)	52,147.85	9.2	−0.398	470
*BraCIPK23.2*		81.02	A09:27469147–27471976 (−strand)	48,712.89	9.14	−0.411	440
*BraCIPK23.3*		73.78	A08:17294577–17297463 (−strand)	44,789.61	8.68	−0.28	400
*BraCIPK24*	AtCIPK24	88.18	A04:9880111–9882974 (−strand)	51,600.66	9.19	−0.267	453
*BraCIPK25*	AtCIPK25	70.14	A06:22263189–22264565 (−strand)	51,839.79	8.68	−0.306	458
*BraCIPK26.1*	AtCIPK26	90.35	A10:14277798–14280847 (−strand)	49,950.04	7.63	−0.502	441
*BraCIPK26.2*		88.08	A02:4468468–4471749 (+strand)	49,780.72	7.65	−0.516	441

**Table 3 genes-13-00795-t003:** Inference of divergence time in paralogous pairs.

Seq 1.	Seq 2	Identity/%	Ks	Ka	ω	T(MYA)
CIPK1.1	CIPK1.2	85.87	0.2033	0.0215	0.105755042	6.7767
CIPK2.2	CIPK2.3	79.05	0.5405	0.0785	0.145235893	18.0167
CIPK4.1	CIPK4.2	44.49	0.5557	0.1255	0.225841281	18.5233
CIPK5	CIPK25	67.54	0.9199	0.0712	0.077399717	30.6633
CIPK6.1	CIPK6.2	90.02	0.5716	0.018	0.031490553	19.0533
CIPK7.1	CIPK7.2	86.49	0.3183	0.0619	0.194470625	10.6100
CIPK9.1	CIPK9.2	89.49	0.3889	0.0302	0.077654924	12.9633
CIPK10.1	CIPK10.2	84.42	0.667	0.0565	0.084707646	22.2333
CIPK11.1	CIPK11.2	82.06	0.6492	0.0437	0.067313617	21.6400
CIPK12.1	CIPK12.2	90.49	0.4008	0.0484	0.120758483	13.3600
CIPK13.2	CIPK13.3	85.69	0.6879	0.0625	0.090856229	22.9300
CIPK17.1	CIPK17.3	54.03	0.1041	0.0042	0.040345821	3.4700
CIPK20.1	CIPK20.2	66.06	0.4802	0.0676	0.140774677	16.0067
CIPK21.1	CIPK21.2	87.74	0.2286	0.046	0.201224847	7.6200
CIPK22.1	CIPK22.2	90.23	0.6802	0.0693	0.101881799	22.6733
CIPK23.1	CIPK23.2	80.38	0.3699	0.0085	0.022979184	12.3300
CIPK26.1	CIPK26.2	86.43	0.1264	0.0681	0.538765823	4.2133
CBL1.1	CBL1.2	96.71	0.4019	0.0092	0.022891266	13.3967
CBL2.1	CBL2.2	84.62	0.4885	0.0093	0.019038	16.2833
CBL3.1	CBL3.3	52.61	0.3695	0.2117	0.572936	12.3167
CBL4.1	CBL4.3	92.31	0.4213	0.0237	0.056254	14.0433
CBL9.1	CBL9.2	98.59	0.3269	0.007	0.021413	10.8967
CBL10.1	CBL10.3	77.11	0.3583	0.0661	0.184482	11.9433

## Data Availability

All data in the present study are available in the public database as referred in the Material and Method part.

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
