# Peer review of "Genome-Wide Identification and Functional Analysis of the Calcineurin B-like Protein and Calcineurin B-like Protein-Interacting Protein Kinase Gene Families in Chinese Cabbage (Brassica rapa ssp. pekinensis)"

_genes, 2022, doi:10.3390/genes13050795_

Round 1
Reviewer 1 Report
Dear Editor,
The manuscript entitled “Genome-Wide Identification and Functional Analysis of the Calcineurin B-like Protein and Calcineurin B-like Protein-Interacting Protein Kinase Gene Families in Chinese Cabbage (Brassica rapa ssp. pekinensis)” was revised. In general, it is a very well prepared manuscript and I think it will be a very useful resource for the literature. I've outlined a few revisions and a few minor recommendations below that I think might have been overlooked. The article needs minor edits.
This revisions described below;
Title
OK
Abstract
Line 24, 25, “Arabidopsis thaliana” should be written in italics
Keywords
should be given in alphabetical order.
Introduction
This is just a suggestion. It would be nice to show the mechanisms with a simple schematic as in the example below for easy understanding. In this case, there is conceptual confusion.
https://www.sciencedirect.com/science/article/pii/S1534580720306298
Line 54-55, “A. thaliana” should be written in italics. Since it is used once in the text before, it is not necessary to write the entire Latin name. It would be appropriate to write it as A. thaliana.
Line 55, “Chlorella vulgaris” …italic…
Line 71, “A. thaliana”
Line 75, “Brassica rapa ssp. pekinensis”
Line 79-81, “The CBL–CIPK …………………………in several species.” ….This sentence needs at least one reference….
Material Methods
Gene symbols are italicized. Symbols for proteins that were named for genes begin with an upper-case letter, but there are no accepted formatting guidelines for proteins that were not named for genes. Protein symbols are not italicized.
https://www.biosciencewriters.com
Line 98, “A. thaliana”…
Line 99, “B. rapa”
Line 108, “A. thaliana, B. rapa”….
Line 131, ………………..200 mmol photons m−2 s−1 light intensity……
Line 131-133, There needs to be a space between some parentheses.
Line 138, Eastep® Super Total RNA Extraction Kit – Promega…………….
Line 146-147, …………. “°C” ………this is the appropriate symbol.
Line 147, “B. rapa”
Results
Line 182, There needs to be a space for brackets
Line 184, 186, “A. thaliana”…
Line 284, “A. thaliana, B. rapa”…. Ä°talics
Line 369, 371, 378, 395, 396, 398, “A. thaliana”…
Discussion
In all studies, including yours, the species Arabidopsis thaliana was used, it would be more appropriate to give it as A. thaliana instead of just the genus name in the text.
Conclusions
OK
Tables
OK
Figures
OK
References
OK

Author Response
Dear Reviewer
Thank you very much for your letter and the comments from the referees about our paper submitted to Genes. (Ms. No. genes-1693230).
We have checked the manuscript and revised it according to the comments. We submit here the revised a list of changes.
If you have any question about this paper, please don’t hesitate to let me know.

Reviewer 2 Report
Regulation of plant responses to abiotic stresses, is a very complex process controlled by various groups of genes. Depending on the signal, these genes can gather and interact with each other as specific sets that trigger a series of defence signals in plants. Due to the changing climatic conditions in the world, plants can adopt specific pathways stimulation systems after the action of a stress factor. One such phenomenon is the ability to produce proteins with different functions or the duplication of genes during evolution.
The presented work describes the possibilities offered by bioinformatics tools, indicating the diversity of gene sequences, protein motifs, etc. The authors indicated and give the insight of the knowledge about the function of genes known from the literature and described as stimulating plant responses to stress. An analysis of a total of 65 genes revealed 3 detected homologs, the functions of which were tested on the A. thaliana model plant.
The experiments in this research were designed with the high level of knowledge of the general topic.
The trait regulation is represented by many different gene transcripts belonging to specific genome regions, which was also indicated by authors using mapping approach studies.
The conducted analyses significantly expand the base of gene sequences related to the regulation of the assessed traits.
Enlarging the knowledge of particular gene groups and homologs and recognition of their activity seems to be the milestone of broaden the knowledge of the full process metabolism.
The manuscript need revise some minor remarks:
How many seeds of Cabbage was used, did they derived from particular plant cross? How many individuals/genotypes where analysed according to mapping studies?
Line 210 Results. With the exception of BraCBL3.3 (looking at the Fig 1 it supposed to be BraCBL3.2, simultaneously in discussion line 428 should be changed.
In the title of figure 3 in the final sentence it supposed to be two instead of three.
Line 309: Results. ………..the treatment, than decreased after 1 hour.
The final paragraph of discussion (from line 473) in my opinion is not necessary and summarize the research as conclusion - for author decision.
Author Response

(The authors gave the same response as above.)
